# Dimerization of the Glucocorticoid Receptor and Its Importance in (Patho)physiology: A Primer

**DOI:** 10.3390/cells11040683

**Published:** 2022-02-15

**Authors:** Steven Timmermans, Jolien Vandewalle, Claude Libert

**Affiliations:** 1Center for Inflammation Research, VIB, 9052 Ghent, Belgium; steven.timmermans@irc.vib-ugent.be (S.T.); jolienvdw@irc.vib-ugent.be (J.V.); 2Department of Biomedical Molecular Biology, Ghent University, 9052 Ghent, Belgium

**Keywords:** glucocorticoids, glucocorticoid receptor, dimerization, sepsis, mutants

## Abstract

The glucocorticoid receptor (GR) is a very versatile protein that comes in several forms, interacts with many proteins and has multiple functions. Numerous therapies are based on GRs’ actions but the occurrence of side effects and reduced responses to glucocorticoids have motivated scientists to study GRs in great detail. The notion that GRs can perform functions as a monomeric protein, but also as a homodimer has raised questions about the underlying mechanisms, structural aspects of dimerization, influencing factors and biological functions. In this review paper, we are providing an overview of the current knowledge and insights about this important aspect of GR biology.

## 1. Introduction

### 1.1. Glucocorticoids

Glucocorticoids (GCs) are steroid hormones produced by the adrenal cortex [1,2]. They have a broad range of effects including developmental, anti-inflammatory [3], metabolic [4,5] and many other functions [2]. GCs are hydrophobic molecules derived from cholesterol and hence they can easily pass cell membranes to enter in cells and exert their effects. GCs were first discovered in 1946 based on research on Addison’s disease, a rare disease, also known as primary adrenal insufficiency or hypoadrenalism [6]. The active form of GCs is known as cortisol in humans and corticosterone in rodents. Owing to their anti-inflammatory and immune-suppressive actions, and small molecular weight, GCs are among the most widely prescribed drugs worldwide and are used both in short-term and chronic settings, despite that their chronic use poses risks for side effects such as osteoporosis [7], or opportunistic infections due to the immune suppressive effects [8]. Furthermore, some patients do not respond to GC treatment and display so-called GC resistance [9], the occurrence of which varies between diseases and can also develop over the course of the treatment [9]. Over the years, much effort has been directed towards developing selective synthetic GCs with fewer side effects while keeping the therapeutic effects intact; however, success has been limited [10,11].

### 1.2. Glucocorticoid Receptor: Structure and Function

GCs bind the GC receptor (GR), also known as Nuclear Receptor 3 C1 (NR3C1). This is a soluble receptor belonging to the superfamily of nuclear receptors (NRs) [2,12]. It is estimated that 1000 to 2000 genes are subject to GR-mediated regulation, and some studies suggest that up to 20% of all genes are responsive to the GR. The GR regulates many pathways (e.g., gluconeogenesis [13,14], inflammatory response [3,15], fatty acid metabolism [16,17]) by regulating gene expression in various organs and tissues (e.g., liver [18,19], nervous system [20,21], adipocytes [17,22]). Over- or under stimulation of the GR results in severe phenotypes such as Cushing syndrome and Addison’s disease [23,24]. The NR family is composed of 49 members and includes other well-known receptors, such as the estrogen receptor (ER) and mineralocorticoid receptor (MR) as well as a number of orphan receptors (the ligands of which are unknown) [25]. Similar to other nuclear receptors, the protein domain configuration of the GR can be subdivided into the following four regions (Figure 1A), from N-terminus to C-terminus [25]: N-terminal domain (NTD), a DNA-binding domain (DBD), a short flexible hinge region and the ligand binding domain (LBD). The N-terminal domain is the least conserved of the four domains. It shows large differences between different nuclear receptors [25] and contains most polymorphisms in humans (mostly without deleterious effect) [26,27]. The NTD is also intrinsically disordered, making it difficult to study its structure, and no crystal structure of this region is available. However, there is some structural information known, mainly secondary structure information. The activator function domain within the NTD is an organized domain in vivo which may adopt variable conformation to induce specific responses by recruiting different cofactors based on its configuration [28]. Expression of the GR-coding gene (*NR3C1*) can start from several alternative transcription initiation sites, which give rise to receptor isoforms differing in the size of the NTD, each with a different transcriptional capability and tissue-specific expression [29,30]. Next to the multiple possible transcription initiation sites, the GR also has multiple possible alternative translation start sites in exon 2, leading to different translational isoforms. Finally, the GR also undergoes alternative splicing, leading to several alternative splice isoforms [31]. These isoforms are formed by splicing variations in the final introns/exons, involving the sequence coding for the LBD and sometimes the hinge region [31]. The main isoforms involve an alternative terminal splice acceptor site, resulting a different final exon used giving the GRα and GRβ isoforms [32,33]. GRα is the canonical active GR while GRβ lacks ligand binding and transactivation activities so that GRα/GRβ and GRβ/GRβ dimers cannot activate gene expression, but can bind DNA. As GRβ is always located in the nucleus, this results in an inhibition of GR activity [33,34]. Without ligand, the GR is kept in the cytoplasm in a multiprotein complex consisting of immunophilins and various chaperones in a configuration optimal for highly sensitive ligand binding [35]. When a ligand binds, the complex changes and recruits other factors [36,37], ultimately resulting in the import of GR into the nucleus through the nuclear pores, where it will exert its regulatory functions [38]. While there is one study that shows GR dimers in cytoplasm, this study was based on a strong overexpression system used in vitro [39]. Furthermore, David Bain’s group showed that GR does not form dimers spontaneously and that an additional factor, such as DNA, is needed [40].

It is generally accepted that GR exerts its function in the nucleus by forming a receptor homodimer and binding DNA at a specific sequence element. Initially it was thought that GR would form tetramers [41], but this was abandoned due to lack of direct evidence and the fact that support was found for the existence of dimers [42]. However, several recent studies, mainly by the group of Diego M. Presman, have once again brought the notion that GR might also exist as a tetramer, or as a dimer of dimers [43,44], to the forefront. The GR dimer model remains the most generally accepted model for GR function and GR tetramerization is an active avenue of research. Next to the dimeric and, and possible tetrameric functions of GR, the GR monomer is also thought to perform several functions. The exact functioning of GR monomers is not yet fully understood, but dimer-deficient mutants have been generated and used to infer monomeric activity. Likewise, some GR ligands that prevent dimer formation, such as compound A, can also be used. Decades ago, it was hypothesized that GR monomers would rather be transcription blockers, either by DNA binding or by inhibitory protein–protein interactions with other transcription factors, such as sequestration and tethering, while GR homodimers would be more responsible for gene upregulation via direct DNA binding. While direct monomeric DNA-binding activities have not been directly observed in a quantitative assay, there is other evidence in support (see further). Evidence for tethering and sequestration of other TFs by GR, which is believed to be a monomeric function, has been found. Since GR is supposed to bind pro-inflammatory transcription factors such as NF-kappaB as a monomer [45], and since GR likely induces expression of some genes involved in gluconeogenesis (Phosphoenolpyruvate carboxykinase 1 (*PCK1*) and Glucose 6-phosphate (*G6PC*), potentially leading to type 2 diabetes mellitus, an important side effect of GCs), as a dimer [14], this led to the paradigm that beneficial, anti-inflammatory, effects were monomer mediated and that undesirable side effects were dimer mediated. It was generally believed that skewing GR into a monomeric form or into a dimeric form could determine therapeutic outcome versus side effects profiles [46]. A balance in favor of monomers would be ideal for chronical treatment, with a minimum of side effects (from GR dimers) of long-term GR activation [47], and stronger dimerization would be better in acute settings (Figure 2) [48]. Indeed, several GR dependent gene products–thought to be dimer dependent–have been identified to have anti-inflammatory functions that can be responsible for the acute anti-inflammatory effects of GCs, as will be illustrated below.

Work has been carried out to search for ligands that push the GR towards monomer or dimer action as respectively Selective GR Activators and Modulators (SEGRAM), such as compound A [49], and Selective Dimerizing Glucocorticoid Receptor Agonists and Modulators (SEDIGRAM) [10]. While several effective synthetic GCs have been developed, so far this approach has not resulted in such “skewing” ligands at the bedside. Dissociating the side effects and anti-inflammatory effects of GCs solely on the basis of their monomeric or dimeric structure turned out to be unrealistic. Indeed, it has been shown that in an acute inflammatory setting, consensus GRE (dimer or higher order) gene expression is required for GC/GR effectiveness. Moreover, some side effects are not mediated via dimeric GR (see Table 1). A good understanding of the mechanisms and functions of receptor dimer formation is needed to help developing such ligands in a more educated way. In this paper, we will provide an overview of the current understanding of GR dimerization, with a focus on insights into dimerization mechanisms and the importance of intact GR dimers in pathophysiological conditions inferred from studies with mutant mice.

## 2. Ligand-Induced GR Homodimer Formation

Both GR monomers and GR dimers are presumed to be able to bind to certain DNA sequences (see later) and to influence transcription. Based on genome-wide ChIP-seq and ChIP-exo studies, it has been shown that GR dimers, and possibly other higher order structures, mediate the transcriptional response to high (i.e., pharmaceutical) doses of GCs [83]. The GR-GRE-mediated gene induction from a proximal promoter is the most straightforward way by which GR dimers lead to gene expression, after recruitment of cofactors and RNA-polymerase complexes. GR-DNA binding and receptor dimerization require involvement of the two Zn-fingers of the DBD. It has been a long-standing point of discussion whether GR oligomerizes before binding DNA or if it occurs as a part of the DNA binding process, especially in vivo. Studies have shown support for both views, coordinated binding of dimers and cooperative binding of monomers [84,85,86,87,88], as well as direct binding to consensus dimer sites (GREs) as a dimer [39,56,57], with possibility of monomeric interactions at half-sites [83]. Furthermore, recent studies have also proposed the possibility of higher order oligomers being formed and contributing to GR transcriptional functions, such as tetramers [43,44], which may possibly extend the aforementioned theories with binding to DNA as dimers followed by tetramer formation.

Structural studies on human, mouse and rat receptors as well as extensive mutagenesis programs have provided much information about the dimerization interfaces. However, due to the presence of disordered regions in the complete receptor, a crystal structure of full length GR has not been obtained so far, but the DBD and LBD domains have been crystalized separately by multiple groups, with 73 structures existing at this time that are publicly available via the protein data bank (PDB) [89]. The majority of the structures describe the DBD in complex with DNA (37/73), but structures from LBD with ligand and/or cofactors and DBD and LBD (closest to full length GR structures, i.e., excluding the NTD) are also available.

### 2.1. The DBD Interface

The GR DBD (from rat GR) was first crystalized in a complex with DNA in 1991 [90]. This structure revealed that the DBD is comprised of two zinc fingers. The first one (mostly N-terminal) recognizes and binds to the DNA with residues in the proximal (P)-box being especially important for function. Sequence specificity is obtained from K442 and V443 residues in the P-box and from the first arginine (R447) downstream of the P-box (Figure 1A). These residues are required for binding the GRE motif and discriminating it from other, related motifs (such as the ER binding site). Other residues can also make further stabilizing contacts, the number and strength of which depend on the exact sequence being bound. The second zinc finger provides a dimerization interface partly overlapping with the distal box (D-box) or D-loop [90,91,92,93] (Figure 1B). In between both Zn-fingers we find a region called ‘the lever arm’, a sequence which appears to transmit subtle structural changes in GR, which are imposed by the nature of the sequence that is bound (so called allosteric effect) (Figure 3). This allows sequenced-based signals to be transmitted and for site specific regulatory events to occur, as shown by work with the GRγ, which has an insertion (Arg) in the lever arm, binds the same sites as GRα, but shows a different regulation profile [94,95,96].

The DBD contacts the DNA in such a way that two GR molecules are oriented in a head-to-head orientation [90,93]. The amino acids in the P-box make contact with the DNA through the major groove [97] and it is the residues K442, V443 and R447 that make contact with specific DNA residues, namely G2 (hydrogen bond with K442), T4′ (van der Waals interaction with V443), G5′ (bidentate hydrogen bonds with R447) from the GRE half-site A_1_G_2_A_3_A_4_C_5_A_6_ [98]. Three of the five amino acids in the D-loop are considered vital to stabilization of the dimer interface: A458, R460 and D462, and in addition there is one residue outside the D-loop that is involved in DBD dimer formation, namely I464 [62,64,66]. Each of these residues on one monomer interacts with the complementary residue on the other monomer, meaning that following interactions are formed: A458–I464′ (backbone hydrogen bond), R460–D462′ (salt bridge) and D462–R460′ (salt bridge) (Figure 1B). The structural information supports that DBD dimerization occurs in a DNA dependent manner, which was also shown by Meijsing et al. [98].

### 2.2. The LBD Interface

Next to the dimerization interface in the DBD, a second dimerization interface in the LBD region was discovered (Figure 4). The structure of GR’s LBD was characterized in depth and the second dimer interface was described by Bledsoe et al. in 2002 [99]. The configuration of this dimer interface was found to be unique to the GR and was not recovered in any other NR, not even in the very closely related MR [25,99]. The LBD of GR consists of 12 alpha-helices (H1 to H12) and four beta-sheets that form a three-layer helical domain consisting of the ligand binding pocket, dimerization domain and activator function domain. The dimerization domain is formed by distant residues brought in proximity by the ternary structure. Hydrophobic interactions and formation of hydrogen bonds are the driving force for stabilization of the LBD dimer interface. The reciprocal interactions between residues P625 and I628, found in the loop between H5-H6, and four hydrogen bonds from the residues between 546 and 551 (between H1 and H3) and Q615 (H5) from both monomers form the core of the dimerization interface. Subsequent analysis [100] using 20 new crystal structures of the LBD found that the LBD dimer interface was less conserved than the DBD dimer interface, as there were fewer conserved residues. This has led to the description of several alternate structures, where other residues are marked as more important than those found in the initial analysis of the LBD dimerization interface. The structure found by Bledsoe et al. was in fact the least often found in structural studies, only six times [101,102,103,104,105], and was the least stable; however, energetically more stable configurations involved much fewer conserved residues. The study by Bianchetti et al. [100] also identified several residues being important that were also found in the Bledsoe structure: Y545, P625, I628 and Q630. Taken together, ligand binding stimulates LBD domain dimerization. Whether dimerization at the level of LBD transmits a signal towards DNA binding or to dimerization at the DBD interphase is not known.

## 3. GR Dimer Mutations

Mutagenesis has played an important role in understanding the function of GR and its relation to structure. Many receptor mutants, ranging from large deletions to point mutants have been created [106]. To study the receptor dimerization mechanism and the effects of GR dimerization on GR function, dimer-deficient receptors have been developed. Most of the GR dimerization mutants have targeted the DBD, mostly because it was the earliest structure resolved and dimerization interphase discovered.

The best characterized and most used of the DBD dimerization mutants is the A458T mutant (human, A465T in mouse, A477T in rat) called the GR^dim^ [46,97]. This mutation was first reported in in vitro studies where GR was mutated and GR-DNA binding and transcriptional activation were studied. This study revealed the importance of the DBD and of several amino acids in this domain, the alanine being recovered as a key amino acid [107]. In the WT GR protein, a hydrogen bond is formed between the backbones of the A458 and the I464′ on the dimer partner. The Ala to Thr substitution in the GR^dim^ version prevents the formation of this interaction [46,59,63].

Other DBD dimer mutants that have been described are the R460D and D462R. These residues form salt bridges with their dimer partners (R460-D462′ and D462-R460′) [90,108]. Each of the two mutations prevent the salt bridge from forming and disrupt the dimer interface. More recently, a multiple mutant was generated and called the GR^dim4^, based on the human GR sequence. This receptor has four mutations, three in the D-loop and one in the lever arm: A458T, R460D, D462C and N454D, and was used for research in U-2 OS cell lines [77].

The effectivity of the GR dimer mutants to address GR monomeric versus GR dimeric functionalities depends on whether these mutants are indeed (completely) dimer deficient. This is, however, a point of controversy. Several studies indeed demonstrate an almost complete loss of dimerization in the GR^dim^ mutant, whilst other studies show almost no difference in dimerization with the WT receptor. The GR^dim^ mutant is used the most to study GR monomer vs dimer functionalities with the assumption that this mutant is indeed completely dimer deficient. However, multiple studies have sought to elucidate the effective oligomerization state of the GR^dim^ mutant. There are several studies that indicate that the GR^dim^ is still able to oligomerize and that the difference between the dimerization level of the WT receptor and the GR^dim^ mutant is small, with GR^dim^ only dimerizing slightly less [77,109]. Indeed, Presman et al. revealed that the A458T mutant is still capable of forming tetrameric structures, and is not hampered by the DBD dimer domain mutation [110]. On the other hand, there are several other studies showing that dimerization in the GR^dim^ mutant is nearly completely absent. Single cell quantitative approaches have shown a strongly reduced dimerization of GR^dim^ in the presence of the ligand, whereas the unliganded GR^dim^ have similar dissociation constants of dimerization as the WT receptor [56,57,79]. We recently confirmed these results using a proximity ligation assay, where we found that the GR^dim^ mutant does not show significant complex formation upon DEX administration; however, we cannot conclude that no dimerization occurs at all, as there is still some GR^dim^ mediated gene expression observed, as has been illustrated before in the literature [111]. In any case, complex formation in GR^dim^ was found to be below the detection limit of the assay, while for GR^wt^ dimers, complex formation could be clearly detected. The controversy about the dimerization capacity of the GR^dim^ mutant might be explained by the different experimental setups applied (assay used, overexpression vs physiological conditions, cell type, in vitro vs in vivo, ligand type, ligand dose, full GR vs only DBD, etc.). In our work determining GR^dim^ dimer formation, we worked in physiological conditions with primary MEF cells derived from mice embryos without overexpression of any GR variant [111]. Similarly, Lim et al. detect little, if any, dimeric occupancy in GR^dim^ mice indicating that this mutant probably interacts with the genome primarily as a monomer under physiological conditions [83]. Many other experiments, including those still showing dimerization, are performed in overexpression systems where the GR of interest is overexpressed with a plasmid [109,110]. As the assays used to evaluate dimerization mostly rely on proximity of molecules as a proxy for complex formation, and as DNA binding itself also plays a role in dimer formation, a different explanation for the observations found in GR^dim^ also needs to be considered. The GR molecules bind to DNA in a cooperative manner [112], and it has also been proven that the GR^dim^ mutant shows a reduction in this cooperativity [94,113]. As a consequence, the GR-DNA complex will form with greater difficulty, and will be less stable which may result in overall less GR molecules in close proximity for an extended period of time, thereby reducing proximity signals and also gene expression. This alteration of function may also help explain the few sites that show gain of function in the GR^dim^ mutant, where expression of associated genes is increased compared to WT.

After the discovery of the LBD dimer interface [99], efforts were made to disrupt dimerization at this location as well, by targeted mutagenesis. This was first reported by mutating position I628 to an alanine. This variant was combined with the A458T mutant in the DBD in order to obtain a receptor that would be completely dimer free and was named GR^mon^ [109]. This new combined mutant indeed displayed the worst ligand-induced dimerization efficiency compared to GR^wt^ and GR^dim^ as evaluated with the Number and Brightness assay, but there was still some dimerization observed at sufficiently high concentrations of DEX [109].

Generation of mutant mice expressing the I628A mutation as well as the double point mutation (A458T and I628A in humans; A465T and I634A in mice) has revealed that both are lethal, because the I628A mutation undermines decent ligand binding. This was recently proven by means of a labeled ligand titration assay [111]. In this study, we demonstrate that GR^wt^ and GR^dim^ have an equivalent affinity for ligands, but the I628A mutation has a strongly reduced ligand affinity [111]. Using high ligand concentrations (≥0.1 µM DEX), the large amount of ligands forces the equilibrium to the bound state and nuclear translocation occurs as normal. However, under low ligand concentration, mimicking physiological levels of GCs (≤10 nM DEX), I628A mutants show almost no ligand binding and a strongly impaired nuclear translocation [111]. The use of the GR^mon^ mutant for the study of GR dimerization thus has a strong confounding factor of reduced ligand binding, which should be considered when interpreting results obtained with this model. Although this finding was unexpected, since this particular amino acid is not directly involved in ligand binding per se, it illustrates that amino acids in the ligand binding pocket (i) are extremely conserved for other reasons than direct ligand binding, e.g., structural aspects, (ii) may have multiple functions e.g., in ligand binding, structure and dimerization and (iii) are therefore never retrieved as viable mutations in the human population.

## 4. GR Dimer and Monomer Transcriptional Regulation

Various mechanisms by which the GR regulates transcription have been described [114]. (i) The term transactivation (literally a contraction of ‘transcriptional activation’) mostly refers to direct transcriptional activity performed by binding of homodimers to a GC response element (GRE) in the DNA. (ii) Binding of GR on composite elements, which contain a (half) GRE and a(n) (overlapping) response element of another TF, for example as GR/MR heterodimers [115], and leads to either gene activation or repression. (iii) Indirect binding (‘tethering’) or transcriptional activation due to half-site binding of the monomeric form of the receptor [83]. Transrepression (transcriptional repression) is achieved by means of tethering and/or sequestration of other, mostly pro-inflammatory, transcription factors [116] or potentially by binding to negative GRE, a class of which was described by Surjit et al. [117].

The classical way for activating GR gene expression is thus the binding of the GR to a GRE sequence in the DNA, which is a motif composed of a hexamer inverted repeat, with both hexamers separated by a three-base spacer and with the consensus sequence of AGAACA(N)_3_TGTTCT [90,118]. On this motif GR binds as a dimer, where the monomers act in a cooperative manner [112]. Genome-wide chromatin immunoprecipitation studies (ChIP-seq and ChIP-exo) of GR in mouse tissues have illustrated that without administration of exogenous ligand (but in the presence of physiological doses of adrenal-produced corticosterone) several genomic sites are occupied [83]. It was shown that these sites primarily contain a GRE-half site, AGAACA, and might thus be bound by monomeric GR [119]. However, it cannot be excluded that binding to this site prevents dimer formation, as a second GR partner may still be recruited. A follow-up study to Shiller et al. may show if any such process occurs. Upon administration of a high dose of ligand, these half-sites appear to be vacated and the monomers are recuperated as GR dimers binding on full GRE motifs. The GR^dim^ mutant has been shown to bind the GRE element with lower affinity than the GR^wt^, because the monomers of the GR^dim^ mutant have a reduced cooperativity when binding DNA as mentioned before [61] and/or because the GR^dim^ is deficient in complex formation as recently shown [79]. Interestingly, it has been demonstrated in vitro that GR^wt^ is fully responsive, the GR^dim^ mutant has a reduced transcriptional response, and the GR^mon^ mutant does not respond upon administration of GCs [120]. Of note, the GR^dim^ mutant can also induce a very small number of genes that are specifically induced or even better induced compared to GR^wt^, indicating a limited gain of function for some genes next to a more general loss of function for most other genes. We could confirm these data with the addition that GR^dim2^ (i.e., I628A alone) performs almost as well as GR^wt^ with 1 µM DEX, and similar to GR^mon^ with 10 nM DEX [111]. Thus, the results from the GR^mon^ with 100 nM DEX [109] may also suffer from the reduced ligand affinity of the I628A mutant as mentioned above. Furthermore, only a small fraction of GREs can be directly linked to gene expression. Some genes do indeed have several GREs, which work together in order to regulate expression. Many GRE sites are more related to enhancer activity then direct transcriptional induction and have been found to modulate the expression level of genes under the control of enhancer regions [121,122].

A second class of elements that can be recognized and bound are the negative GREs (nGRE). This nGRE element was described very early in GR research based on its function: a DNA element that can induce GR-mediated repression. These elements differ from consensus GREs and show little sequence conservation among them [123]. A large class of more conserved nGREs is described by Surjit et al. [117]. These inverted repeat nGRE (IR-nGRE) motifs are characterized by the consensus sequence CTCC(N)_0–2_GGAGA [117], which provide one high-affinity (CTCC) and one low-affinity (GGAGA) half-site. Unlike a consensus GRE, the IR-nGRE is not bound by GR dimers, but by monomers in a tail-to-tail manner that prevents contact between the D-loops of both partners [124]. Binding of one monomer on the high-affinity half-site hinders binding of the other monomer, resulting in negative cooperativity [124], which may result in low binding affinities making sites low responsive. However, recent work studying intestinal epithelial cells (IEC) in GR^dim/dim^ mice found that the repression of Stat1 was reduced in these mice [59]. Indeed, the GR^dim^ mutation A458T substantially decreases the overall binding of its DBD to the DNA, mainly to the GRE, but also binding to IR-nGRE is negatively affected [124]. The existence of these IR-nGRE elements is controversial, some groups have provided evidence for IR-nGRE activity [59,125], while others could not reproducibly identify them on a genome-wide basis [126]. In case of the Stat1 promoter, GR was recruited twofold more to the IR-nGRE elements of Stat1 upon DEX treatment in GR^wt/wt^ mice compared with DEX-treated GR^dim/dim^ mice [59]. While direct binding to the actual IR-nGRE is not shown, the regions were selected based on the presence of the IR-nGRE sequence. Further work investigating the presence (and function) of any type of nGRE elements could be important in fully understanding GR function and clearing up the current questions about (IR-)nGREs. Importantly, the other single mutants that were generated for the DBD dimerization domain (at R460 and D462) display only a reduction in GRE binding, but still effectively bind the IR-nGRE motif [124]. These data point to the fact that the popular GR^dim^ mutant may also impact some monomeric functions of the GR and that the other mutants may thus provide some additional information in comparison to the GR^dim^ mutant. Furthermore, the reduced repression of Stat1 in GR^dim^ mice can also be explained by other mechanisms, such as microRNA-based regulation of Stat1 in a GR dimer dependent manner. Indeed, GR dimer regulated transactivation can regulate the expression of several micro-RNAs, as has been shown for mmu-miR-511 which subsequently downregulates TNFR1 expression [127].

The GR is also capable of indirect binding to DNA by protein–protein interaction with another protein, mostly a TF. When performing GR ChIP-seq analysis, this is often recovered as regions with high GR signal, but no GRE site [83,128]. A variety of other TF motifs are recovered in these regions (e.g., [128]). This mechanism is usually related to GR’s transrepression properties where pro-inflammatory TFs are bound by GR and so prevent the inflammatory TFs from transactivating their target genes. The monomeric form of GR is thought to be responsible for these effects [120,129]. However, other evidence has shown that transrepression and transactivation cannot be assigned so clearly to monomer and dimer receptor actions [48]. This will be discussed in more detail below with examples where dimeric GR is believed to be required to survive acute inflammatory diseases via both gene activating and gene repressing properties inferred from studies using the GR^dim/dim^ mutant.

## 5. Alternative Partners for Dimer Formation

The bulk of GR functions are performed by either the monomer or the homodimer form of the receptors. Nevertheless, GR can form heterodimers, particularly with other NRs yielding active TF complexes in which GR teams up with and forms a heterodimer to regulate specific genes. The formation of such heterodimers is a less explored field. The GR/MR heterodimer is the most studied, because the MR binds and is activated by endogenous and some synthetic GCs [130,131]. MR/GR heterodimers have mainly been studied in the central nervous system [108,132]. One study demonstrated that MR/GR dimers, as well as GR/GR dimers, are formed in the cytoplasm and that alternative dimer interfaces are involved [39]. Specifically, the hinge region of the GR was found to be vital for GR/GR and GR/MR dimerization in solution (without DNA) and the LBD of the GR was involved in the formation of GR/MR heterodimers [39]. A recent study confirmed the GR/MR interaction in the nucleus in vivo [115].

Several older studies also suggest heterodimer formation with the androgen receptor [133]. Recently, there has been increased interest in this GR/AR interaction, as it might open up new therapeutic avenues for disease treatment [134]. Both receptors can recognize repeats of TGTTCT, and a 2018 study reported that androgens can modulate GR activity in liver and adipose tissue [135,136,137]. While the existence of GR/AR dimers is not proven, they are considered possible based on observed effects and high similarity of the LBD [105,106,110].

The interaction between GR and PPARα has also been a subject of interest. These have been shown to be able to interact in vitro, but this has not been proven in vivo [138,139]. Evidence from DNA binding of GR and PPARα allows for the possibility of GR/PPARα dimer formation, especially considering that it was shown that PPARα is required for proper GR function [140], but also for other mechanisms, such as PPARα ligand synthesis by GR target genes [134]. Overall research efforts in finding non-classical dimers, such as GR heterodimers is limited and may provide a new avenue for medical research with yet untapped possibilities.

## 6. Role of GR Complex Formation in SIRS and Sepsis

### 6.1. SIRS and Sepsis

GR homodimerization is believed to be important for different physiological and pathological functions of GR, which are summarized in Table 1 (see also ref [93]). This table summarizes the findings using the GR^dim/dim^ mice. Given the recent insights contributing to our understanding of the hypersensitivity of GR^dim/dim^ mice in systemic inflammatory response syndrome (SIRS) and sepsis, we will focus our discussion on this topic. The different phenotypes observed in GR^dim/dim^ mice during acute inflammation illustrates the pleiotropic mechanisms of how dimeric GR is thought to control inflammation.

SIRS is characterized by a fast, systemic release of cytokines, such as tumor necrosis factor (TNF), interferons (IFNs), interleukin 6 (IL-6) and IL-1β, as a response to a noxious stressor such as trauma or ischemia. Sepsis evolves from an infection that causes life-threatening organ dysfunction resulting from a dysregulated host response. In contrast to SIRS, sepsis involves activation of both pro- and anti-inflammatory responses, along with abnormalities in non-immune compartments, such as the cardiovascular, metabolic and coagulation compartments [141]. According to the latest global estimates of sepsis incidence and mortality, 49 million people are affected yearly, leading to 11 million deaths, corresponding to 20% of all deaths worldwide [142]. Injection of TNF or lipopolysaccharides (LPS), the latter being cell wall components of Gram-negative bacteria, are animal models used as models for SIRS. One of the most relevant models for sepsis is the cecal ligation and puncture (CLP) model. In this model, the cecum is ligated and punctured using a needle, followed by resuscitation with antibiotic-containing fluids [143]. That GCs protect against sterile SIRS, was shown many years ago, by applying the TNF and LPS models [144,145,146].

### 6.2. Anti-Inflammatory Genes Induced by GR Complex Formation

GR^dim/dim^ mice are extremely sensitive to TNF-induced SIRS [58,59]. Mortality rate is significantly higher in GR^dim/dim^ mice compared to their WT counterparts, and this is associated with higher plasma IL-6 levels and more severe intestinal damage [58,59]. Mitogen-activated protein kinase phosphatase *1* (MKP-1) plays a key role herein. MKP-1 is induced upon dexamethasone (DEX) or TNF injection in GR^wt/wt^ mice and not in GR^dim/dim^ mice as a consequence of reduced binding of the mutant GR to the GRE of the MKP-1 coding gene *Dusp1*. Dusp1^−/−^ mice are similarly sensitized for TNF as GR^dim/dim^ mice and show increased levels of phosphorylated Jun N-terminal kinases (JNK), which promote apoptosis in liver tissue. Loss of *Jnk2* partially rescues the TNF hypersensitivity of Dusp1^–/–^ and GR^dim/dim^ mice. These data illustrate the important role of GR complex formation in resisting TNF-induced SIRS through inhibiting JNK2 activation via MKP-1 activation [58]. Another anti-inflammatory gene requiring an intact GR dimerization profile is Sphingosine kinase 1 (*SphK1* encoding S1P). This gene is synergistically induced by GCs and pro-inflammatory stimuli via the GR in macrophages, resulting in increased circulation of S1P during inflammation. GR^dim/dim^ macrophages lack binding of the mutant GR to the GRE of *SphK1* resulting in reduced S1P release in GR^dim/dim^ mice upon LPS treatment. S1P is essential to limit lung inflammation induced by LPS endotoxemia as DEX can no longer protect against acute lung injury in absence of GR or SphK1 in myeloid cells, or in GR^dim/dim^ mice [60]. A last example illustrating the importance of GR complex formation to protect against sepsis is *Tsc22d3* encoding Glucocorticoid Induced Leucine Zipper (GILZ). Full expression of *Tsc22d3* requires an intact GR dimerization potential [59,83]. GILZ is typically induced by GCs, but upon inflammation this gene is reduced in several cell types, such as hepatocytes and blood cells [147]. However, the mechanism behind the downregulation of GILZ in inflammatory settings is not yet understood. A possible explanation could be that due to the GC unresponsiveness in sepsis [65], the GR is unresponsive towards GR’s endogenous ligand cortisol/corticosterone, thereby leading to decreased expression of this gene. The recently registered RECORDS trial is therefore using GILZ expression in blood as a marker of corticosteroid activity for the rapid recognition of GC resistance in sepsis patients (NCT04280497). However, it should be noted that other mechanisms, next to GR-mediated transcription, can control levels of GILZ. For example, it has been illustrated that the RNA-binding protein tristetraprolin can reduce GILZ mRNA stability upon TLR activation [148]. Mice with GILZ overexpression (GILZ-tg mice) have a reduced mortality towards CLP-induced peritonitis, which could be linked to an enhanced bacterial clearance [147]. Moreover, overexpression of GILZ specifically in monocytes and macrophages similarly reduced mortality rates in the CLP model as the full GILZ-transgenic mice, and this was also associated with an increased bacterial clearance due to enhanced phagocytosis capacity of macrophages [149]. Taken together, the data suppose that anti-inflammatory genes requiring an intact GR dimerization potential (i.e., *Dusp1*, *SphK1* and *Tsc22d3*) are essential to transmit the protective effects of GR in SIRS and sepsis.

### 6.3. Pro-Inflammatory Genes Suppressed by GR Complex Formation

In addition to its typical transactivation potential, GR dimers may downregulate hundreds of genes by interaction with IR-nGRE [117]. For example, based on studies using GR^dim/dim^ mice, GR dimers are supposed to directly bind to IR-nGRE elements in the STAT1 promoter resulting in reduced STAT1 induction and phosphorylation. Unchallenged GR^dim/dim^ mice, which lack the potential to bind these short DNA sequences, subsequently present a strong IFN-stimulated gene (ISG) signature. This ISG signature is gut-specific and dependent on the gut microbiota as assessed with antibiotics studies. Injection of TNF in GR^dim/dim^ mice leads to an even more outspoken induction of these ISGs, including necroptosis master switches *Ripk3*, *Zbp1* and *Mlkl*, compared to their WT counterparts [59]. The increased expression of these genes contributes to necroptotic cell death in the intestinal epithelial cells (IECs) and subsequent intestinal leakage. DEX is able to protect GR^wt/wt^ mice challenged with a lethal TNF dose, whereas this is not the case for GR^dim/dim^ mice challenged with their respective lethal TNF dose. The transcription factor binding motifs found in GR^wt/wt^-specific DEX downregulated genes were specifically ISRE and IRF elements. As such, DEX has a lower impact on the repression on TNF-induced ISGs and concomitant intestinal damage in GR^dim/dim^ compared to GR^wt/wt^ mice. Nec1s, a specific necroptosis inhibitor partially rescues the hypersensitivity of GR^dim/dim^ mice to TNF, thereby illustrating the essential role for GR complex formation in resisting TNF-induced SIRS trough inhibiting necroptosis in the intestinal epithelium [59].

Next to the observed effects of the GR^dim/dim^ mutant in IECs, this mutation also has clear effects on macrophage function. In contrast to TNF, expression of IL-1β is prolonged in GR^dim/dim^ mice that succumb to LPS-induced shock [64]. This may be attributed to the role of GR dimerization in macrophages, as BMDMs derived from GR^dim/dim^ mice are refractory to GC treatment as assessed by the production of IL-1β upon addition of LPS and/or DEX. Inhibiting IL-1β using an IL-1 receptor antagonist partially rescues the hypersensitivity of GR^dim/dim^ mice to LPS, thereby illustrating an important role for complex formation in resisting LPS-induced mortality through modulating IL-1β production [64]. Interestingly, GR^lysmKO^ mice, which lack the GR in their myeloid cells, similarly show increased susceptibility towards LPS injection, but IL-1β inhibition completely protects in this mouse model. Since GR^dim/dim^ mice carry the point mutation in all cell types, these data suggest that GR dimerization in other cell types also contributes to survival during sepsis, and moreover, that the monomeric function of GR in myeloid cells also provides some protection [64]. Indeed, we recently showed that GCs apply two key mechanisms to control endotoxemia. One at the level of macrophages, where GCs are thought to suppress TNF production in a GR monomer-dependent way, and one at the level of the intestinal epithelium, where GCs are believed to suppress TNFR1-induced ISG gene expression and necroptotic cell death in a dimer-dependent way [61]. Lastly, Osteopontin (OPN), a crucial mediator for lung inflammation, is increased in the lungs of GR^dim/dim^ mice challenged with LPS compared to GR^wt/wt^ mice. A partial role for GR complex formation in macrophages was found herein, as BMDMs derived from GR^dim/dim^ mice showed a trend towards induced *Opn* upon LPS treatment, compared to their WT counterparts [62]. However, as this induction was not significant, a possible role for GR in other cell types contributing to *Opn* regulation cannot be excluded and remains to be elucidated. Collectively, these data demonstrate that suppression of pro-inflammatory genes in an assumedly GR dimer-dependent way (*Stat-1*, *IL-1β* and *Opn*) are essential for protection against SIRS and sepsis.

### 6.4. Hemodynamic and Metabolic Parameters Controlled by GR Complex Formation

Next to the direct effect of GR dimerization on the inflammatory compartment of sepsis, GR dimerization is also thought to be protective in sepsis by controlling hemodynamics and metabolic parameters. For example, GR^dim/dim^ mice challenged with LPS have a compromised systemic hemodynamic stability and require increased norepinephrine to maintain hemodynamic stability [62]. Moreover, LPS treatment in GR^dim/dim^ mice leads to increased levels of lactate when compared to WT mice. The higher lactate levels might be linked to a disturbed mitochondrial function in the heart of GR^dim/dim^ animals [62]. Another clarification for the increased lactate levels of GR^dim/dim^ mice is via reduced clearance of lactate. Indeed, the main mechanism to clear lactate is via the Cori cycle, i.e., conversion of lactate to glucose in the liver via gluconeogenesis and release of glucose in the blood which then can be taken up by peripheral tissues. Gene expression profiling of livers of prednisolone-treated GR^wt/wt^ and GR^dim/dim^ mice, identified reduced fold changes of typical gluconeogenic genes (e.g., *Pck1*, *Irs1*) in GR^dim/dim^ mice compared to GR^wt/wt^ mice, thereby suggesting a role for GR dimerization in regulating gluconeogenesis [67]. Indeed, GR^dim/dim^ mice show more pronounced hypoglycemia after LPS challenge compared to GR^wt/wt^ mice [150]. Interestingly, next to the role of GR dimerization in controlling glucose and lactate levels, administration of lactate leading to peak blood lactate values of 20 mM, which is the value of lactate typically observed in the sickest mice after CLP surgery, does not cause detrimental effects in GR^wt/wt^ mice, whereas it does lead to acute lethality in GR^dim/dim^ mice [65]. This lethality could be linked to an uncontrolled production of vascular endothelial growth factor (VEGF), resulting in vascular leakage, severe hypotension and organ damage [65]. How and where GR dimerization is supposed to control lactate toxicity remains to be studied. One possibility is through regulating lactate-induced VEGF production, for example in macrophages. Another possibility is by protecting the barrier function of the endothelium. Indeed, deletion of GR specifically in the endothelium renders mice hypersensitive for endotoxemia [151]. Whether this protective effect of GR in the endothelium is monomer- or rather dimer-dependent has not been evaluated.

Taken together, GR complex formation is important in multiple cell types (i.a., intestinal epithelium and macrophages) to protect against SIRS and sepsis (Figure 5). This protection is believed to depend on dimeric regulation of both pro- and anti-inflammatory genes through binding respectively IR-nGRE and GRE sequences. Next to their basal increased susceptibility towards SIRS and sepsis, GR^dim/dim^ mice are also refractory to GC treatment in SIRS conditions. Moreover, GR complex formation also controls critical hemodynamic and metabolic parameters essential for surviving acute diseases such as SIRS and sepsis.

## 7. Conclusions

Historically, it has been considered that the beneficial, anti-inflammatory effects of GCs are mediated through the monomeric GR, whereas its side-effects are mainly caused by dimeric GR. This has prompted research into the development of various tools to study the monomeric and dimeric functions of GR and into the design of drugs with improved therapeutic index, namely SEGRAMs. The GR^dim^ mutant and the GR^dim/dim^ mouse model has been instrumental in that regard. While there were doubts about how completely the dimer formation is disrupted from the start, and while recent work has shown that the GR^dim^ does indeed still form dimers, it is generally accepted that the balance of the GR monomer/dimer abundance is very clearly and very strongly shifted towards monomers. After the discovery of a second dimerization domain in the LBD, a second GR mutant was made that combines the A458T (GR^dim^) and I628A (LBD dimerization interface) mutation to get a completely dimer-dead receptor, called the GR^mon^ mutant. This receptor does indeed not form any complex, but based on data from the LBD structure, the affinity for ligand might also be negatively affected. This adds a strong confounding effect to the GR^mon^ mutant unrelated to dimer formation. Thus, the GR^dim^ mutant remains very useful, also because an in vivo model exists for this mutant. Furthermore, recent work has shown that the beneficial, anti-inflammatory effects of GCs and their unwanted side effects cannot be distinguished so clearly based on the respectively monomeric and dimeric actions of GR. Especially in acute settings, such as SIRS and sepsis, the GR^dim/dim^ mutant has clearly reduced response to GCs. Moreover, in contrast to the known function of dimeric GR inferred from the GR^dim/dim^ mutant to induce transcription of anti-inflammatory genes, such as *Dusp1*, *SphK1* and *Tsc22d3*, GR dimers are also believed to be essential in suppressing pro-inflammatory gene transcription through binding, for example, the IR-nGRE elements in the STAT1 promoter. Lastly, more research is warranted to understand how exactly complex formation is involved in regulating the hemodynamic and metabolic abnormalities that are observed in SIRS and sepsis. Taken together, despite GCs already having been used for more than 70 years in the clinic, we remain surprised by the ingenuity of the different mechanisms and actions exerted by GCs. More research will definitely provide us with more insights in how to generate GCs with improved therapeutic index.

## Figures and Tables

**Figure 1 cells-11-00683-f001:**
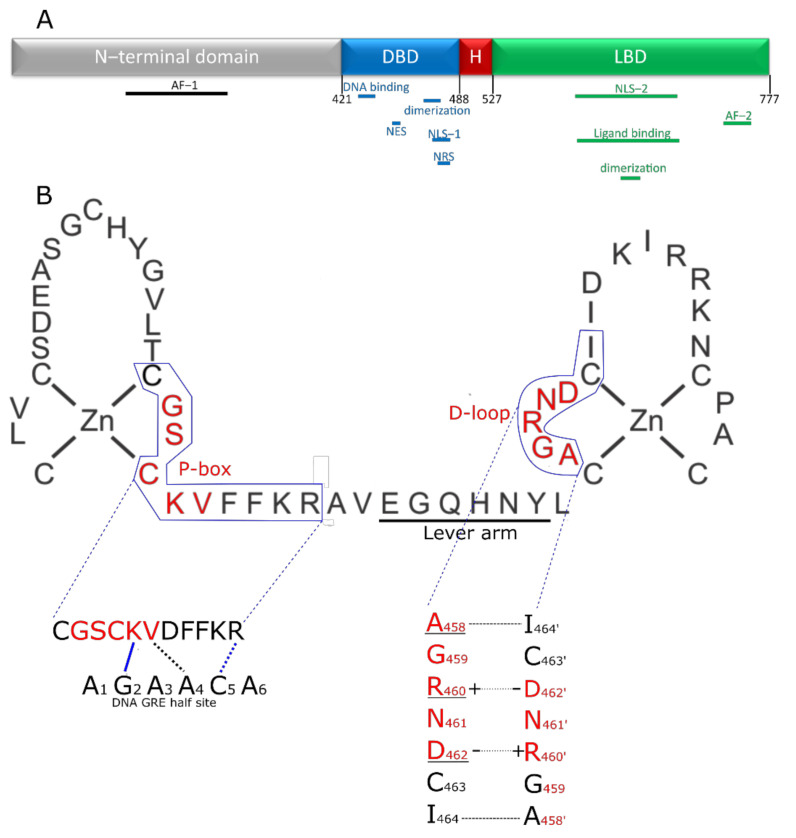
Structure of the GR and its DBD. (**A**) Domain structure of the GR. The disordered N-terminal domain contains one of the two activator functions (AF-1) for receptor transcriptional activity. The DNA binding domain (DBD; blue) contains the zinc finger element responsible for recognizing and binding the GR response element in the DNA and a second zinc finger providing the primary dimerization interface between two GR molecules. In addition, the DBD contains nuclear localization (NLS-1), nuclear retention (NRS) and nuclear export signal (NES) peptide sequences. The hinge region (H; red) is a short flexible linker connecting the DBD to the C-terminal ligand binding domain (LBD). This LBD is responsible for binding ligands in the ligand binding pocket and also contains a nuclear localization signal (NLS-2) and a second dimerization interface. Finally, the second, and most powerful, activation function (AF-2) is also located in the LBD. (**B**) The first zinc finger of the DBD plays a role in recognizing and binding the GRE, through the P-box. The second zinc finger is responsible for homodimerization of GR. The residues in the D-loop are especially important for this functionality. The lever arm connects both zinc fingers and changes conformation depending on the exact DNA sequence bound, transmitting DNA sequence information to the rest of the receptor. Interaction of the DBD of 1 GR partner with the DNA is depicted, with P-box shown in red. Blue lines indicate hydrogen bond interactions, black lines indicate van der Waals interactions, dotted lines indicates that the interaction is with the complementary nucleotide of the opposing strand. The K and V residues in the P-box make site specific contacts with the DNA and recognize G2 and T_4′_ residues respectively. The arginine outside the p-box interacts with G_5′_. These 3 GR-DNA interactions are of great importance, but depending on sequence context, other residues also make further stabilizing contacts with the DNA. Dimer stabilizing interaction between 2 GR molecules is depicted in the lower left figure. A458 makes a backbone-based hydrogen bond contact with I464′ (and I464 with A458′). The interface is further stabilized by 2 salt bridges formed by R460 and D462′ and D462 and R460′. Underlined residues have been subjected to mutagenesis to yield GR dimer deficient mutants, of which A458T, GR^dim^, has been used the most in scientific research, with many publications studying it directly (e.g., if it still forms dimers), or using it as a poorly dimerizing receptor.

**Figure 2 cells-11-00683-f002:**
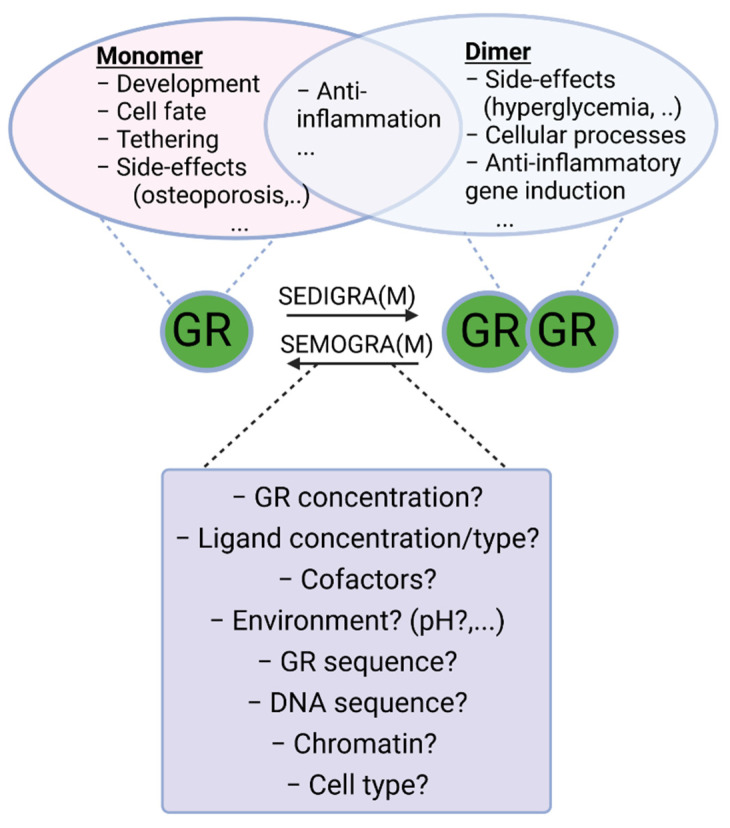
Monomeric and dimeric functions of GR. Work has been carried out to search for ligands that push the GR towards monomeric or dimeric action for chronic and acute inflammatory diseases, respectively. Indeed, both forms of GR possess anti-inflammatory actions trough tethering and anti-inflammatory gene induction, respectively. Most of the side-effects of GCs (hyperglycemia, glucocorticoid resistance, etc.) are ascribed to its dimeric functions, and therefore SEMOGRAMs are believed to improve the therapeutic index in chronic settings requiring long-term use of GCs. In acute inflammatory diseases, such as SIRS and sepsis, however, dimeric GR is believed to be essential to limit inflammation and therefore SEDIGRAMs are favored in these settings. A good understanding of the mechanisms determining the balance between GR monomers and dimers is needed in the search for such dissociating ligands. SEMOGRAM: selective monomerizing GR agonists and modulators, SEDIGRAM: selective dimerizing GR agonists or modulators.

**Figure 3 cells-11-00683-f003:**
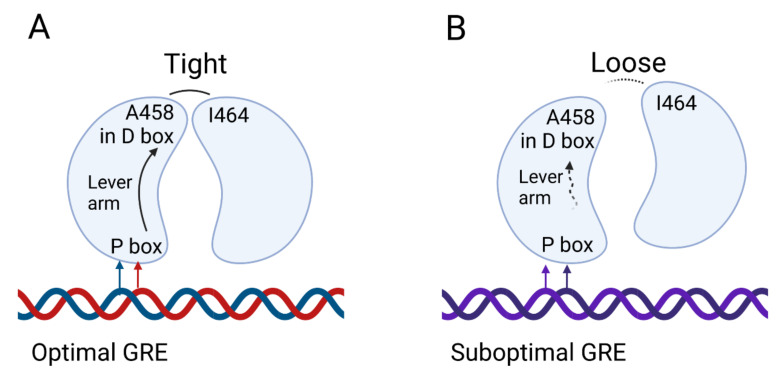
DNA binding controls GR dimerization at the level of the DBD. Binding of two DBD domains from two GR molecules onto DNA is depicted. (**A**) Upon binding of the first DBD/GR monomer to an optimal GRE, the DNA sequence directs the conformation of the lever arm. The lever arm subsequently translates this information to the D box in the DBD and allows interaction between A458 in the first monomer and I464 in the second monomer. This tight interaction stabilizes the second monomer on the other half-site and induces gene transcription. (**B**) In case of a suboptimal GRE, suboptimal stabilization occurs, as not all auxiliary GR-DNA contacts can be formed. Furthermore, it is possible that the GR structure is thereby affected through the lever arm which is sensitive to the sequence context, and that the D-boxes are affected reducing DNA-dependent dimer formation and cooperativity.

**Figure 4 cells-11-00683-f004:**
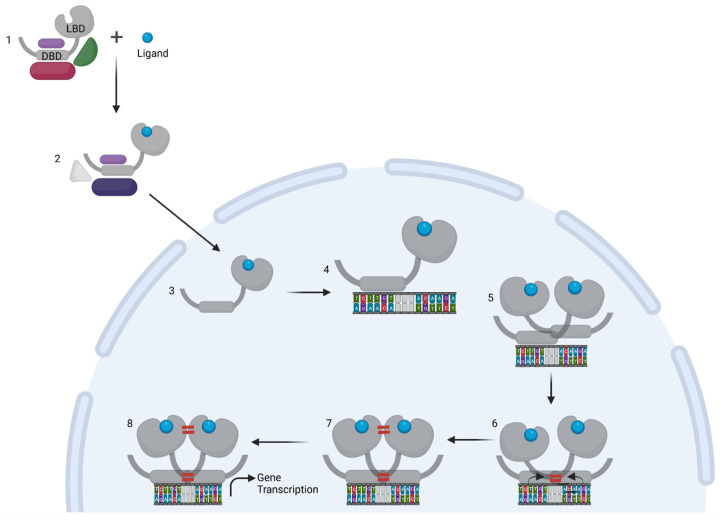
Overview illustrating the process of GR dimerization resulting in gene transcription. (1) Under unstimulated conditions, the GR is found in the cytoplasm as part of a chaperone protein complex. (2) Upon binding of the ligand, the complex undergoes changes that eventually lead to (3) the import of GR into the nucleus. (4–5) For dimer-mediated action, the GR binds DNA on its motif, AGAACAnnnTGTTCT, with each half-site binding one GR monomer. (6) The DNA binding causes changes in the lever arm of the DBD leading to DBD dimer interface stabilization. (7) The proximity of both LBDs brings their dimerization interfaces in close contact allowing a second dimer interaction to form. (8) The DNA-bound dimer is then fully formed, with maximal stabilizing contacts between DBD and LBD dimer interfaces and can now recruit cofactors to modulate transcription of target genes.

**Figure 5 cells-11-00683-f005:**
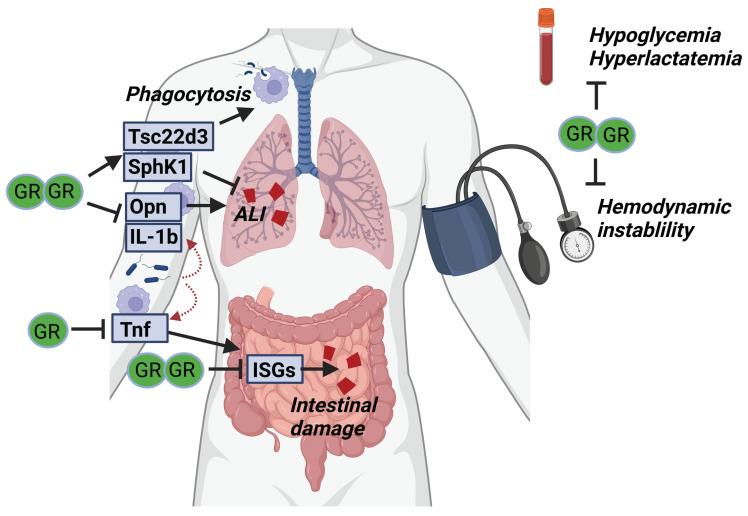
GR complex formation is required for surviving SIRS and sepsis. Upon infection, pro-inflammatory cytokines (Opn, IL-1b and TNF) are produced by myeloid cells and cause acute lung injury (ALI) and intestinal damage. GR dimerization is thought to be necessary to control these pro-inflammatory genes and prevent organ damage. In addition, anti-inflammatory genes (*Tsc22d3*, *SphK1*) are produced by the myeloid cells, likely in a dimer-dependent way, which is necessary to further suppress organ damage. Next to the effects of GR dimerization on the inflammatory compartment, the dimeric GR may be essential to prevent hemodynamic instability and control glucose and lactate levels during sepsis.

**Table 1 cells-11-00683-t001:** Phenotypes retrieved from GR^dim/dim^ mice in different physiological and pathological processes.

Process	Effect in GR^dim/dim^ Mutant	References
Resolution of inflammation		
Antigen- and G6PI-induced arthritis	DEX protection lost	[50]
Serum transfer-induced arthritis	DEX protection lost	[51]
Contact hypersensitivity	DEX protection lost	[52]
PMA-induced irritative skin inflammation	DEX protection intact	[53,54]
Experimental autoimmune encephalomyelitis	DEX protection intact	[55]
Allergic airway inflammation	DEX protection lost	[56]
Graft- vs host disease	Increased mortality	[57]
TNF-induced SIRS	Increased mortality + DEX protection lost	[58,59]
LPS-induced SIRS	Increased mortality + DEX protection lost	[60,61,62,63]
CLP-induced septic shock	Increased mortality	[64,65]
Side effects		
Hyperglycemia	Pred effect reduced	[66,67]
Osteoporosis	Pred/DEX effect intact	[68,69,70]
Skeletal muscle atrophy	DEX effect intact	[71]
Wound repair	Wound repair reduced	[72]
Gastroparesis and gastric acid secretion	DEX effect lost	[73]
Ocular hypertension leading to glaucoma	DEX effect lost	[74]
Glucocorticoid resistance	DEX effect lost	[75]
Cellular processes		
Adipogenesis	No adipogenesis	[76]
Apoptosis	DEX effect lost	[46,77]
Proliferation	Proliferation reduced	[46]
Spatial memory	Spatial memory reduced	[78]
Cognitive function under stress condition	CORT effect reduced	[79]
Weight control	Body weight increased	[80]
Activation HPA axis in 6% hypoxia	Activation of HPA axis reduced	[81]
Trauma-induced fracture healing	Protected	[82]

Abbreviations: G6PI; glucose-6-phosphate isomerase, DEX; dexamethasone, Pred; prednisolone, PMA; phorbol myristate acetate, TNF; tumor necrosis factor, LPS; lipopolysaccharide, CLP; cecal ligation and puncture, SIRS; systemic inflammatory response syndrome, CORT; corticosterone, HPA; hypothalamic-pituitary-adrenal.

## Data Availability

Not applicable.

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
