# Peer review of "Dimerization of the Glucocorticoid Receptor and Its Importance in (Patho)physiology: A Primer"

_cells, 2022, doi:10.3390/cells11040683_

Round 1

Reviewer 1 Report

This manuscript wades into the controversial area of the separate functions of GR as a monomer and dimer in cells and physiologically.  There is a summarization of what is known about GR dimerization, though this summary is not complete without the inclusion of work by David Bain.  The paper largely reinforces the dogma that monomeric functions of GR are beneficial, while dimeric GR functions can result in side effects particularly pharmaceutically. Although the manuscript delves into the stoichiometry of GR, and discusses some limitations of the dim mutant, it nonetheless bases much of the discussion of monomeric GR function on results using the dim mutant.  For the reasons in the annotated PDF attached and the comments below, I think this is misleading and muddies the waters of understanding how regulation of GR stoichiometry affects its function.

My main issue with the paper is the characterization of the dim mutant of GR functioning as a monomer. The evidence that GR does anything strictly as a monomer has never been shown, it has been inferred.  The work of the Hager lab indicates that the dim mutant still exists primarily as a dimer and tetramer.  The work of Schiller and Yamamoto show that the dim mutant more frequently binds to a sequence containing a half site than wt, but still binds dimeric sites, and does not exclude the possibility that GR still functions from these sites as a dimer.  The biochemistry of the dim mutant (Meijsing, Watson) suggests that it is less cooperative than wt, and yet it still binds to sites as a dimer. The binding and function of GR at nGREs is highly controversial, and has only been shown by the Chambon and Ortlund lab and has not been reproducible in many other labs.  The blanket characterization of dim, while discussed in this manuscript, as functioning as a monomer is simply not supported well by data.  

In addition, exceptional work has been done by David Bain to show that, in the absence of other factors in vitro, full-length GR is primarily a monomer. This suggests that cellular factors aid GR in dimerizing in the cell.  In the case of DNA binding, the factor is DNA which induces a structural change in the DBD to allow cooperative binding,  But other factors may play a role off DNA.

The most accurate model of the dim mutant is that it is impaired in cooperative DNA binding, and that the mutation changes GR function, including potentially some gain of function at genes.  Those two aspects of this mutant may not always be linked.  Thus, the bulk of the manuscript linking a requirement for dimerization to certain functions or that GRdim functions as a monomer are suspect.  Accordingly, I think that this review can be misleading about how stoichiometry affects GR function. 

There are other issues about the interpretation of data in the annotated PDF, and the manuscript organization could use shortening and editing for consistency and redundancy. 

Reviewer 2 Report

See attached file

Round 2

Reviewer 2 Report

none

Author Response

There are no new comments from rev 2.